# Impact of *UGT1A1* Polymorphisms on Febrile Neutropenia in Pancreatic Cancer Patients Receiving FOLFIRINOX: A Single-Center Cohort Study

**DOI:** 10.3390/cancers14051244

**Published:** 2022-02-28

**Authors:** Jiyoung Keum, Hee Seung Lee, Jung Hyun Jo, Moon Jae Chung, Jeong Youp Park, Seung Woo Park, Si Young Song, Seungmin Bang

**Affiliations:** 1Division of Gastroenterology, Department of Internal Medicine, Institute of Gastroenterology, Yonsei University College of Medicine, Seoul 03722, Korea; 01212@eumc.ac.kr (J.K.); lhs6865@yuhs.ac (H.S.L.); junghyunjo83@yuhs.ac (J.H.J.); mjchung@yuhs.ac (M.J.C.); sensass@yuhs.ac (J.Y.P.); swoopark@yuhs.ac (S.W.P.); sysong@yuhs.ac (S.Y.S.); 2Division of Gastroenterology, Department of Internal Medicine, Ewha Womans University College of Medicine, Seoul 07985, Korea

**Keywords:** pancreatic cancer, FOLFIRINOX, *UGT1A1*, febrile neutropenia, neutropenia

## Abstract

**Simple Summary:**

FOLFIRINOX, which is a first-line chemotherapy for metastatic pancreatic cancer, has become one of the high-risk regimens related to developing febrile neutropenia (FN). Although *UGT1A1* polymorphisms are associated with the metabolism of irinotecan, their role as surrogate markers for FOLFIRINOX-induced neutropenia has not been confirmed. In this retrospective study, a total of 154 patients (FN group (*n* = 31) vs. non-FN group (*n* = 123)) were divided into three groups based on the predicted *UGT1A1* phenotype (extensive metabolizer (EM) vs. intermediate metabolizer (IM) vs. poor metabolizer (PM)). The Cox regression analysis showed that female sex (hazard ratio (HR): 2.20; *p* = 0.031), ECOG PS = 1 (HR: 2.83; *p* = 0.008), *UGT1A1* IM (HR: 4.30; *p* = 0.004), and *UGT1A1* PM (HR: 4.03; *p* = 0.028) were independent risk factor of FN. We propose *UGT1A1* as the strongest predictive factor for FN and the need for *UGT1A1* screening prior to chemotherapy.

**Abstract:**

FOLFIRINOX (oxaliplatin, leucovorin, irinotecan, and 5-fluorouracil) is a first-line chemotherapy for metastatic pancreatic cancer (PC). Chemotherapy-induced neutropenia is one of the most serious adverse events associated with advanced PC. Although *UGT1A1* polymorphisms are associated with the metabolism of irinotecan, their role as surrogate markers for FOLFIRINOX-induced neutropenia has not been confirmed. We investigated risk factors for FN—in particular, *UGT1A1* polymorphisms—in PC patients receiving FOLFIRINOX, using a single-center cohort registry. To investigate the association between *UGT1A1* polymorphisms and FN, we divided patients into three groups based on the predicted *UGT1A1* phenotype: extensive metabolizer (EM) vs. intermediate metabolizer (IM) vs. poor metabolizer (PM). A total of 154 patients (FN group (*n* = 31) vs. non-FN group (*n* = 123)) receiving first-line FOLFIRINOX were identified between December 2017 and July 2020. The Cox regression analysis showed that female sex (HR: 2.20; *p* = 0.031), Eastern Cooperative Oncology Group performance status = 1 (HR: 2.83; *p* = 0.008), *UGT1A1* IM (HR: 4.30; *p* = 0.004), and *UGT1A1* PM (HR: 4.03; *p* = 0.028) were statistically significant risk factors for FN. We propose that *UGT1A1* is the strongest predictive factor for FN and that this gene should be screened prior to the administration of chemotherapy.

## 1. Introduction

Pancreatic cancer (PC) is a lethal malignant neoplasm, and surgical resection is the only curative treatment. Unfortunately, less than 20% of newly diagnosed PC patients are appropriate candidates for surgical resection [1]. Most other PC patients are diagnosed at an advanced stage and receive chemotherapy.

The FOLFIRINOX regimen, which consists of oxaliplatin, leucovorin, irinotecan, and 5-fluorouracil (5-FU), showed remarkably improved overall survival and objective tumor response rates in the PRODIGE4/ACCORD11 trial [2]. FOLFIRINOX has since become a first-line chemotherapeutic for metastatic PC [3,4].

However, the FOLFIRINOX regimen has unresolved problems regarding dose modification of chemotherapeutic agents and the use of prophylactic granulocyte colony-stimulating factor (G-CSF) in relation to hematological adverse events (AEs), such as febrile neutropenia (FN) and severe neutropenia (NP). Previous studies reported that grade 3/4 NP and FN developed in 45.7–77.8% and 5.4–22.2% of patients, respectively [2,5,6]. In particular, hematological AEs were more commonly seen in the Asian population than in the Western population [5,7,8]. In clinical practice, such a high probability of AEs is a reason for concern when using FOLFIRINOX. Based on the National Comprehensive Cancer Network (NCCN) guidelines version 4.2021, FOLFIRINOX has become one of the high-risk regimens related to the development of FN [9]. In this revised guideline, G-CSF is not routinely recommended as a primary prophylaxis but can be considered in patients with high-risk clinical characteristics.

There have been few studies on the high-risk population for FN occurrence in patients with PC receiving FOLFIRINOX. Two retrospective studies have reported female sex, overweight, initial biliary stent insertion, platelet count ≤ 15 × 10^4^/L, and heterozygosity for either the *UGT1A1*28* or *UGT1A1*6* alleles as risk factors for FN in PC patients receiving FOLFIRINOX [10,11]. Irinotecan, one of the principal drugs of the FOLFIRINOX regimen, is a topoisomerase I inhibitor widely used in the treatment of cancer [12,13]. However, irinotecan presents a high incidence of toxicity, particularly severe NP, and diarrhea [14,15,16]. Irinotecan is converted to an active metabolite known as SN-38, subsequently inactivated and detoxified to an inactive form, SN-38 glucuronide (SN-38G), by a uridine diphosphate-glucuronosyl-transferase (UGT) enzyme encoded by the *UGT1A1* gene [17]. The risk of irinotecan toxicity increases with the presence of genetic variants related to decreased UGT enzyme activity, including *UGT1A1*28* and *UGT1A1*6* [18,19,20]. These variants result in decreased excretion of irinotecan metabolites, resulting in increased active irinotecan metabolites in the blood [21]. According to the Clinical Pharmacogenetics Implementation Consortium (CPIC) guidelines, which provide genotype-guided pharmacotherapy recommendations, classify the *UGT1A1* phenotype as follows: extensive metabolizer (EM), intermediate metabolizer (IM), and poor metabolizer (PM) [22,23]. Individuals who are heterozygous for a reduced function allele (e.g., having a *UGT1A1**1/*28 genotype) are expected to be IMs, and those who are carriers of two reduced function alleles (e.g., having a *UGT1A1**28/*28 genotype) are expected to be PMs. The US Food and Drug Administration (FDA)-approved product label for irinotecan recommends a reduced dose for patients homozygous for the *UGT1A1*28* allele; however, the specific dose reduction for this patient population is not described. Because of uncertainty about clinical usefulness, a preemptive *UGT1A1* test is not widely accepted.

Previous studies have not shown the impact of *UGT1A1* polymorphisms, including the *UGT1A1* PM phenotype, on FN development in PC patients receiving FOLFIRINOX. Therefore, in the present study, we investigated the risk factors for FN—in particular, *UGT1A1* polymorphisms—in PC patients receiving FOLFIRINOX.

## 2. Materials and Methods

### 2.1. Study Design

We retrospectively reviewed the data of all patients diagnosed with PC, using the pancreatic cancer cohort registry of Severance Hospital, which is a prospective database for patients with PC treated with chemotherapy since 2015 [24]. 

The eligibility criteria were as follows: (1) ≥18 years of age; (2) histologically or cytologically confirmed pancreatic adenocarcinoma; (3) at least one measurable or evaluable lesion following the Response Evaluation Criteria in Solid Tumors (RECIST), version 1.1 [25]; (4) Eastern Cooperative Oncology Group (ECOG) performance status (PS) ≤ 1; (5) no previous anticancer treatment; (6) no FOLFIIRNOX dose reduction at the beginning; (7) no prophylactic pegfilgrastim or G-CSF; (7) available *UGT1A1* genotype; and (8) adequate organ function (absolute neutrophil count [ANC] ≥ 1500 cells/μL, creatinine clearance ≥ 50 mL/min).

A total of 495 patients with PC who were treated with first line FOLFIRINOX between December 2017 and October 2020 were ascertained. We identified 154 patients who met the eligibility criteria. This study was approved by the Yonsei University Health System Institutional Review Board (approval number: 4-2020-1060) and conducted according to the principles of the Declaration of Helsinki. 

### 2.2. Chemotherapy Schedule and Response Evaluation

The FOLFIRINOX regimen consisted of oxaliplatin (85 mg/m^2^) as a 2-h intravenous infusion (IVF), followed by leucovorin (400 mg/m^2^) administered as a 2-h IVF, and after 30 min, the addition of irinotecan (180 mg/m^2^) given as 90-min IVF, immediately followed by a 5-FU 400 mg/m^2^ bolus and 2400 mg/m^2^ IVF for 46 h, every 2 weeks.

At the start of treatment, the following tumor-related factors were studied and recorded: patient demographics; body mass index; date of diagnosis; tumor location; cancer stage; laboratory data, including levels of carbohydrate antigen (CA) 19-9; and *UGT1A1* genotypes. Chemotherapy was postponed, or the dose was modified depending on the physician’s decision, and treatment continued until the disease progressed, the toxicity was deemed to be unacceptable, or the patient refused treatment.

To assess treatment efficacy, computed tomography (CT) and serum CA 19-9 tests were conducted every 8 weeks. Treatment responses under the RECIST criteria were recorded by designated radiologists, and the attending physicians independently made the final judgment regarding treatment response. 

### 2.3. Assessment of Chemotherapy-Related Adverse Events

To monitor for treatment-related AEs, the occurrence of AEs was thoroughly assessed by physicians and registered nurses at each visit during the course of chemotherapy. The severity grade of the AEs was recorded in the patients’ medical records according to the Common Terminology Criteria for Adverse Events (CTCAE) version 5.0 [26]. The incidence of AEs was evaluated during the chemotherapy. The data were collected on 13 May 2021.

### 2.4. Febrile Neutropenia and Grade 4 Neutropenia

We defined FN as a single temperature of ≥38.3 °C (101 °F) or a temperature of ≥38.0 °C (100.4 °F) sustained over an hour and an ANC ≤ 500 cells/μL, as mentioned by the Infectious Disease Society of America (IDSA, 2010) guidelines [27]. Grade 4 neutropenia (NP grade 4) is defined as an ANC < 500 cells/μL according to CTCAE, version 5.0.

### 2.5. UGT1A1 Polymorphisms

To investigate the association between *UGT1A1* polymorphisms and FN development, patients were divided into three groups based on the predicted *UGT1A1* phenotype. An individual who has a heterozygous genotype for one decreased function allele (e.g., *UGT1A1**1/*28 or *UGT1A1**1/*6) was classified as an IM. A carrier of two decreased function alleles (e.g., *UGT1A1**28/*28, *UGT1A1**6/*6, or *UGT1A1**6/*28 genotypes) was classified as a PM [23]. An individual carrying two normal function alleles (e.g., a *UGT1A1**1/*1 genotype) was classified as an EM. All patients in this study were initially administered the original dose of FOLFIRINOX, regardless of the *UGT1A1* genotype.

### 2.6. Overall Survival and Progression-Free Survival 

The date of death and last follow-up were reviewed to estimate overall survival (OS) and progression-free survival (PFS). Survival and follow-up data were recorded until 13 May 2021. The OS was computed from the date of diagnosis to the date of the last follow-up or death. PFS was calculated from the date of diagnosis to disease progression (or last follow-up or death). Living patients whose disease did not progress were censored at the date of the last follow-up.

### 2.7. Statistical Analyses

Baseline patient characteristics, laboratory data, treatment characteristics, and frequency of AEs were used to calculate descriptive statistics. The differences in baseline characteristics between the FN and non-FN groups were analyzed by using the Chi-squared test or Fisher’s exact test for categorical variables, and the Student’s *t*-test or Mann–Whitney test for continuous variables. For the comparison of the three groups according to *UGT1A1* polymorphism, the Kruskal–Wallis test was performed, followed by the Mann–Whitney test with Bonferroni correction. A logistic progression model was used to estimate the odds ratio (OR) of potential risk factors for the occurrence of FN or NP grade 4. We calculated the median OS and PFS according to FN, using Kaplan–Meier curves, and compared these by using the log-rank test. Estimated medians with 95% confidence intervals (CIs) are reported. The Cox proportional hazard model was used to estimate the hazard ratio (HR) for FN or NP grade 4 events for each variable. Statistical significance was set at a two-tailed *p*-value of less than 0.05 for all tests. All analyses were conducted by using SPSS version 23.0 (SPSS, Chicago, IL, USA). 

## 3. Results

### 3.1. Patient Characteristics

A total of 154 patients (31 in the FN group vs. 123 in the non-FN group) met the eligibility criteria. The baseline characteristics of the patients are shown in Table 1. The median age of the patients was 62 years (interquartile range: 55–67 years), and 95 of the 154 patients (61.7%) were male. A total of 55 of the 154 patients (35.7%) had locally advanced PC, and 63 of the 154 (40.9%) had metastatic PC at diagnosis. The other 36 of the 154 patients (23.4%) were composed of 12 patients (7.8%) with resectable PC, and 24 patients (15.6%) with borderline resectable PC. Above them, eight (66.7%) patients of resectable PC, and 16 (66.7%) patients of borderline resectable PC were received surgical resection for curative aim. Of the 154 patients, 71 (46.1%) were EMs, 66 (42.9%) were IMs, and 17 (11.0%) were PMs according to their *UGT1A1* genotypes (Appendix A). Among the IMs, 31.2% (48/154) had a *UGT1A1**1/*6 genotype and 11.7% (18/54) had a *UGT1A1**1/*28 genotype. Among the PMs, the *UGT1A1**6/*28 genotype was the most prevalent (7.1%, 11/154), followed by the *UGT1A1**6/*6 (3.2%, 5/154) and *UGT1A1**28/*28 genotypes (0.6%, 1/154). FN developed in 31 (20.1%) patients, and NP grade 4 developed in 94 (61.0%) patients; 54.8% (17/31) of the FN and 64.9% (61/94) of the NP grade 4 occurred within four chemotherapy cycles, respectively.

The proportion of female patients, as well as patients with poor ECOG PS, tumor location at the pancreas head, *UGT1A1* IM, *UGT1A1* PM, high platelet count, and total bilirubin level, was significantly higher in the FN group than in the non-FN group. The other variables were not statistically significantly different between the two groups.

The accumulated dose of irinotecan, oxaliplatin, 5-FU bolus, and continuous infusion of four cycles of chemotherapy was not statistically significantly different between the FN and non-FN groups (Table 2). The median time to FN development was the first four cycles. For this reason, cumulative doses in the non-FN group were computed for up to four cycles of chemotherapy.

### 3.2. Risk Factors for the Development of Febrile Neutropenia

Univariate logistic regression analysis was conducted to identify factors associated with FN, and then multivariate analysis was performed (Table 3). In the multivariate analysis, female sex (OR, 2.87; *p* = 0.021), ECOG PS = 1 (OR, 3.86; *p* = 0.015), tumor location of the pancreatic head (OR, 3.06; *p* = 0.023), *UGT1A1* IM group (OR, 4.78; *p* = 0.005), and *UGT1A1* PM group (OR, 4.86; *p* = 0.035) were independently associated with FN. Since the FN events that occurred in a short period of time and those that occurred together with the accumulation of chemotherapy were very different, the time parameters until the time of occurrence must be fully considered. To reduce the time bias, we conducted Cox regression analysis, which showed that female sex (HR, 2.20; *p* = 0.031), ECOG PS = 1 (HR, 2.83; *p* = 0.008), *UGT1A1* IM group (HR 4.30; *p* = 0.004), and *UGT1A1* PM group (HR 4.03; *p* = 0.028) were statistically significant risk factors (Table 4).

### 3.3. Risk Factors for the Development of Grade 4 Neutropenia

Univariate logistic regression analysis was conducted to identify factors associated with NP grade 4, and multivariable analysis was performed (Table 5). In multivariate analysis, female sex (OR, 3.44; *p* = 0.001) and the *UGT1A1* PM group (OR, 4.06; *p* = 0.044) were independently associated with NP grade 4. When the Cox regression analysis was applied with the concept of time, female sex (HR, 1.98; *p* = 0.001) was found to be a significant risk factor (Table 6).

We included *UGT1A1* IM group in the multivariate analysis in spite of statistical insignificance (*p* = 0.231), because previous studies showed that IM group was related to neutropenia.

### 3.4. Relationship between UGT1A1 Phenotypes and Hematologic Toxicities 

We investigated the association between *UGT1A1* phenotypes and serious hematologic AEs (Grade III/IV). Table 7 shows that there were significant differences in the aspects of anemia (EMs 4.2% vs. IMs 19.7% and PMs 23.5%, *p* = 0.011) and FN (EMs 7.0% vs. IMs 31.8% and PMs 29.4%, *p* = 0.001) among the three groups of *UGT1A1* phenotypes. The *UGT1A1* IM and PM groups tended to be susceptible to NP grade 4, although the differences were not statistically significant. 

We further analyzed the impact of *UGT1A1* polymorphisms on diarrhea in Table 8. Neither diarrhea nor severe diarrhea (Grade III/IV) showed a significant association with the *UGT1A1* polymorphism. 

We investigated the baseline characteristics among the three *UGT1A1* phenotype groups, including chemotherapy characteristics, to rule out selection bias (Appendix A). The total bilirubin before chemotherapy was significantly higher in PMs, compared with EMs and IMs (PMs 1.2 vs. EMs 0.5 and IMs 0.6, *p* < 0.001), although the other variables showed no differences. 

### 3.5. Overall Survival and Progression-Free Survival

The median follow-up period was 11.2 months (interquartile range 7.5–15.9), and the median number of chemotherapy cycles per patient was 11 (interquartile range 6.0–18.3). During this period, 43 patients (27.9%) died, and 89 patients (57.8%) experienced disease progression. 

Our analysis of the treatment outcomes according to FN development showed that the FN group exerted a significant unfavorable effect on OS (18.7 months vs. 24.3 months; log-rank *p* = 0.007) than the non-FN group (Figure 1a). Of the 31 patients in the FN group, seven (22.6%) experienced septic shock, and one patient died due to FN-related septic shock. However, the FN group did not have a significant effect on PFS (Figure 1b).

We compared the relative dose intensity (RDI) of FOLFIRINOX between the FN and non-FN groups to determine the reason for the difference in OS. The FN group received a lower dose of the FOLFIRINOX regimen than the non-FN group (Appendix A). The median RDI of irinotecan, oxaliplatin, and 5-FU (infusion) was significantly lower in the FN group. In contrast, the median duration of chemotherapy did not differ between the two groups.

The analysis revealed that the treatment outcomes according to the *UGT1A1* phenotype did not have a significant effect on OS and PFS (Figure 2a,b).

## 4. Discussion

The FOLFIRINOX regimen is often limited because of the high incidence of FN. In this study, FN developed in 31 (20.1%) patients, and NP grade 4 developed in 94 (61.0%) patients. Similarly, previous studies reported FN and grade 3/4 NP in 5.4–22.2% and 45.7–77.8% of patients, respectively [2,5,6]. We identified female sex, ECOG PS = 1, *UGT1A1* IM, and *UGT1A1* PM as risk factors for FN in PC patients receiving FOLFIRINOX. Among them, the HR of *UGT1A1* status was the highest. 

There have been few studies on the impact of *UGT1A1* polymorphisms on FN development in PC patients treated with FOLFIRINOX. Most studies on the interaction of irinotecan with *UGT1A1* have focused on patients with colorectal cancer (CRC) receiving the FOLFIRI (irinotecan, fluorouracil, and leucovorin) regimen [28,29,30,31,32]. Some guidelines recommend genotyping for *UGT1A1* and considering at least a 25% reduction in irinotecan for patients homozygous for *UGT1A1*28* [33,34]. However, *UGT1A1* genotyping is not routinely performed before the start of irinotecan-containing chemotherapy because of the scarcity of large prospective studies [35]. 

FOLFIRINOX, which adds oxaliplatin over the FOLFIRI regimen, should be evaluated for the impact of *UGT1A1* on PC patients with poorer prognosis than CRC patients. In a multicenter observational study of 199 PC patients in Japan, patients heterozygous for *UGT1A1* were reported to suffer more severe diarrhea than those with the wild-type genotype when receiving the original dose of FOLFIRINOX [36]. In a retrospective study of 106 PC patients treated with FOLFIRINOX in a single-center, female sex, overweight, and biliary stent insertion were reported as risk factors for FN development [11]. Since our study did not perform a multivariate analysis of *UGT1A1*, we could only identify risk factors for the clinical characteristics of patients. In a prospective study of patients with advanced gastrointestinal cancer, including PC, *UGT1A1* genotyping-guided modified FOLFIRINOX dose failed to demonstrate tolerability to PC patients [37]. In contrast, in a prospective study of 50 patients with advanced gastrointestinal cancer including 30 patients with PC, *UGT1A1*-genotyping-guided dose modification of FOLFIRABRAX (irinotecan, fluorouracil, leucovorin, and nab-paclitaxel) with prophylactic pegfilgrastim was tolerable in patients with *UGT1A1* wild type or *UGT1A1**1/*28 genotypes. However, this study evaluated the dose-limiting toxicity (DLT) rate in the first cycle only, and the impact on the entire period of chemotherapy was unknown [38].

In this study, we investigated the risk factors for FN development through multivariate analysis, including the genetic factor of *UGT1A1* status, during the entire chemotherapy period after receiving FOLFIRINOX. *UGT1A1* was found to be a more powerful risk factor than the clinical features of patients. To the best of our knowledge, this study is the first to analyze risk factors for FN in PC patients receiving FOLFIRINOX as first-line treatment, including patients’ clinical features and *UGT1A1* status.

In the present study, PMs for the *UGT1A1* gene were found in 11.0% of PC patients, with the *UGT1A1**6/*28 genotype being the most prevalent (7.1%), followed by *UGT1A1**6/*6 (3.2%) and *UGT1A1**28/*28 (0.6%). IMs for the *UGT1A1* gene were found in 42.9% of patients, with the *UGT1A1**1/*6 genotype being the most common (31.2%), followed by *UGT1A1**1/*28 (11.7%). EMs were found in 46.1% of patients. Our data are generally in accordance with those reported by the CPIC guidelines, which showed that the prevalence of PMs, IMs, and EMs is 8%, 42%, and 50%, respectively, in East Asians [22].

*UGT1A1*28* and *UGT1A1*6* are well-studied *UGT1A1* polymorphisms with respect to irinotecan pharmacokinetics and pharmacodynamics. Especially among Caucasian patients, *UGT1A1*28* appears to be a strong predictor of NP. However, in Asian populations, the *UGT1A1*6* variant is more common and seems to be a more precise predictor of NP. [39,40] In this study population, *UGT1A1*6*, with a minor allele frequency of 22.4%, was more common than *UGT1A1*28*, with a minor allele frequency of 10.1%, which is generally consistent with the Korean Reference Genome database [41]. 

Several studies have shown an association between the degree of systemic exposure to SN-38 and the risk of neutropenia. Minami et al. reported the pharmacokinetics of irinotecan in patients with or without *UGT1A1*6* or **28* in 177 Japanese cancer patients, in which the area under the concentration curve ratio of SN-38G to SN-38 was lower in patients with one and two haplotypes harboring *6 or *28 than in those without *6 or *28, and lowest in patients with two haplotypes harboring *6 or *28 [42]. 

Most studies support a “gene–drug exposure” interaction, in which toxicities among *UGT1A1* polymorphism carriers were associated with higher levels of exposure to irinotecan [43]. The highest correlations with severe NP and diarrhea were found in *UGT1A1*28* homozygotes with irinotecan doses of 180 mg/m^2^, especially at doses of 250 mg/m^2^ [34,43,44]. The dose of irinotecan, which may increase the risk of toxicity to *UGT1A1* IMs, has not been fully determined. A trial using genotype-guided dosing of irinotecan in CRC patients receiving FOLFIRI regimen found that the *1/*1 and *1/*28 genotypes could tolerate higher than the recommended 180 mg/m^2^ every 2 weeks [29]. However, many existing studies have specifically investigated CRC, and there are insufficient studies on the toxicity and efficacy of the FOLFIRINOX regimen associated with *UGT1A1* polymorphisms in PC patients.

A retrospective study of 31 PC patients with the *UGT1A1* PM phenotype reported that an initial dose of irinotecan ≤120 mg/m^2^ reduced NP grade 4 (20%; 5/24) more than an initial dose of irinotecan ≥150 mg/m^2^ (67%; 4/6) [45]. A response rate of 21.4%, a median PFS of 8.1 months, and a median OS of 15.8 months in patients given irinotecan at an initial dose of ≤120 mg/m^2^ were not different from previous studies. However, the authors concluded that a prospective study is needed to determine the RDI for validating the AEs and efficacy of FOLFIRINOX for *UGT1A1* PM, because various doses of irinotecan were administered, and each group of patients divided by the different doses of irinotecan rendered only a few cases. In the present study, the median RDI of irinotecan in the FN group was 88% of the standard dose, and the median RDI of the *UGT1A1* PM group was 89%, indicating a higher dose than in previous studies. Future studies are needed to determine the dose of irinotecan according to the high-risk features of FN and *UGT1A1* polymorphisms.

A few studies have started FOLFIRINOX at the original dose for PC patients with *UGT1A1* PM; therefore, the present study is considered to be the first report on the incidence of AEs. Of the 17 patients in the *UGT1A1* PM group, 88.2% experienced serious (grade 3 or 4) NP, and 29.4% experienced FN, far more than in the PRODIGE4/ACCORD11 trial (45.7%, and 5.4%, respectively). Likewise, the incidence of serious anemia and thrombocytopenia was higher than that in a previous study (23.5% vs. 7.8% and 17.6% vs. 9.1%, respectively).

In addition to genetic factors, female sex and ECOG PS = 1 were risk factors for FN development. In our previous study, female sex was an independent risk factor for FN occurrence and ECOG PS = 1 was not statistically significant but showed a tendency for vulnerability to FN, which was generally consistent with the results of this study [11]. Innocenti et al. reported that females tended to be more prone to NP when irinotecan was administered to patients with CRC or lymphoma [16,46]. Although the mechanism by which females are more likely to develop FN has not been elucidated, Crawford et al. suggested that differences in total bone mass might cause the higher incidence of FN observed in females [47] because lower bone-marrow volume might be associated with lower tolerance to chemotherapy. Four studies in patients with lymphoma, breast cancer, and lung cancer have shown poor performance as risk factors for FN development [48]. Additionally, it has been suggested that physiological age, as indicated by ECOG PS, may serve as a risk factor for chronological age in older patients. 

FN often results in severe infections, longer hospital stays, life-threatening morbidity, and mortality. Patients may also lose potential opportunities for treatment because of the severe consequences of FN. In the present study, seven patients (22.6%) in 31 of the FN group experienced septic shock, and one patient died due to FN-related septic shock. This patient had the *UGT1A1**6/*6 genotype. Four patients in six of the non-mortality cases had *UGT1A1**1/*6 genotypes, and two patients had the *UGT1A1**1/*1 genotype. Moreover, there was a significant difference in OS between the FN and non-FN groups (median OS, 18.7 months vs. 24.3 months, log-rank *p* = 0.007). However, there was no difference in median PFS between the two groups. At present, we do not know the precise reason for the difference between OS and PFS. This might be related to the fact that six of the 31 patients in the FN group suffered from severe FN in which chemotherapy was delayed for more than a week, and one died from FN-related infections. A retrospective study of 165 PC patients treated with FOLFIRINOX reported that prophylactic primary G-CSF usage reduced the risk of NP (55.6% to 31.6%; *p* = 0.003) and FN (18.5% to 1.8%; *p* = 0.002) and improved OS (8.8 to 14.7 months; HR: 1.766, *p* = 0.001) [49]. The author explained that an increase in RDI and treatment duration of FOLFIRINOX could result in a positive impact on survival. Similarly, there was a decrease in the median RDI of each chemotherapeutic agent except 5-FU (bolus) in the FN group compared with the non-FN group. However, there was no difference in the median number of chemotherapy cycles between the two groups. The causal relationship between reduced RDI and negative impact on OS could not be assessed due to the retrospective design of this study. It is necessary to confirm this finding in large prospective studies.

Regarding the association between *UGT1A1* and OS or PFS, there was no difference among *UGT1A1* status. No previous study has evaluated the influence of *UGT1A1* polymorphisms on survival in PC patients receiving FOLFIRINOX. CRC patients with one or more *UGT1A1*28* alleles did not show significant results for OS and PFS [30,50]. Conflicting results also exist. In a prospective study, 250 metastatic CRC patients were treated with the FOLFIRI regimen as a first-line treatment. It suggested no significant survival benefit but revealed a positive effect on tumor response in patients with the *UGT1A1* PM phenotype. The higher response rates were explained by higher SN-38 concentrations in these patients [28]. Further studies are needed regarding the association between *UGT1A1* polymorphisms and therapeutic outcomes in PC patients receiving FOLFIRINOX.

This study had some limitations. First, according to previous studies on pharmacokinetics data of irinotecan, *UGT1A1* PMs were expected to experience higher rates of AEs than *UGT1A1* IMs; however, in this study, the HR on FN did not differ from that of IMs. This is possibly because the number of *UGT1A1* PMs was smaller than that of IMs and EMs. Second, additional irinotecan pharmacokinetics data are needed to confirm the negative effect of *UGT1A1* polymorphisms on lower survival in the FN group. Depending on the actual exposure to SN-38, evidence of modified FOLFIRINOX by genotype-guided dosing and the use of prophylactic G-CSF may be established in the high-risk group for FN development. Third, we could not evaluate genetic variants other than *UGT1A1*. It has been suggested that variations in the genes encoding CYP3A4, SLCO1B1, and ABC transporters, as well as other UGT1A enzymes, are related to the pharmacokinetics and pharmacodynamics of irinotecan [22,46]. With respect to fluorouracil, DPD, TYMS, DPYS, CDA, and MTHFR have previously been reported to be associated with toxicity. It should be considered that these polymorphisms exist at lower frequencies among Asians, and the patient population in this study consisted of East Asians [51,52,53]. Since the use of next-generation sequencing and gene panel tests will increase in the future, additional studies on the association of the pharmacokinetics of FOLFIRINOX with variants in genes other than *UGT1A* are needed. Fourth, our results might not be applicable to the Western population, because *UGT1A1*6* is predominantly found in the Asian population. In our study, among the seven serious FN events, five patients had more than one *UGT1A1*6* allele. Of the 31 FN cases, 21 patients had one or more *UGT1A1*6* alleles, while only seven patients had one or more *UGT1A1*28* alleles. Fifth, for the application of real-world practice, it is necessary to evaluate the toxicity and efficacy of *UGT1A1* genotype-guided dosing of FOLFIRINOX through a large-scale prospective study. Sixth, in this study, neither diarrhea nor severe diarrhea (grade III/IV) demonstrated a significant association with *UGT1A1* polymorphism. It differed from earlier studies that reported that more diarrhea in the *UGT1A1* IM and PM groups, compared to the EM group, was probably the result of the small sample size. Retrospective analysis could also influence this outcome. Seventh, we tried to analyze the association of recent surgery (≤1 month before chemotherapy start) and FN. However, of 24 patients who underwent surgery, only one of them was indicated as having had recent surgery, so an analysis could not be performed. Recently, FOLFIRINOX has been administered as adjuvant chemotherapy for PC patients after surgery, so further research on the association between surgery and FN is needed.

## 5. Conclusions

We investigated the polyfactorial basis of FN, a common adverse effect of the FOLFIRINOX regimen. To our knowledge, a comprehensive investigation such as this has never been conducted, and we propose that *UGT1A1* is the strongest predictive factor for FN and that this gene should be screened prior to chemotherapy. Further studies are necessary to establish definitive guidelines for the dose reduction of FOLFIRINOX and prophylactic G-CSF usage in PC patients with *UGT1A1* polymorphisms. 

## Figures and Tables

**Figure 1 cancers-14-01244-f001:**
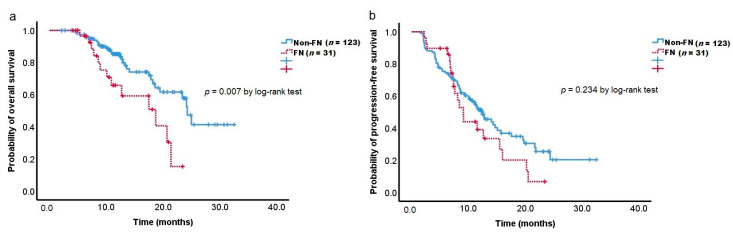
Kaplan–Meier estimates of overall survival (**a**) and progression-free survival (**b**) according to febrile neutropenia.

**Figure 2 cancers-14-01244-f002:**
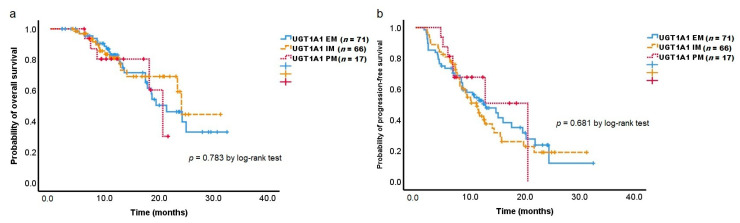
Kaplan–Meier estimates of overall survival (**a**) and progression-free survival (**b**) according to *UGT1A1* phenotype.

**Table 1 cancers-14-01244-t001:** Baseline characteristics.

Variable		All Patients (*n =* 154)	FN Group (*n =* 31)	Non-FN Group (*n =* 123)	*p*-Value
Patient characteristics					
Age, years		62 (55–67)	66 (62–70)	60 (55–66)	<0.001
Sex, no. (%)					
Male		95 (61.7)	13 (41.9)	82 (66.7)	0.011
Female		59 (38.3)	18 (58.1)	41 (33.3)	
ECOG PS					
0		130 (84.4)	21 (67.7)	109 (88.6)	0.010
1		24 (15.6)	10 (32.3)	14 (11.4)	
BMI, median (kg/m^2^)		22.7 (21.2–25.0)	23.6 (21.2–26.1)	22.3 (21.2–24.9)	0.499
DM		50 (32.5)	13 (41.9)	37 (30.1)	0.208
*UGT1A1*					
Extensive metabolizer		71 (46.1)	5 (16.1)	66 (53.7)	<0.001
Intermediate metabolizer		66 (42.9)	21 (67.7)	45 (36.6)	
Poor metabolizer		17 (11.0)	5 (16.1)	12 (9.8)	
Tumor characteristics					
Location					
Head		83 (53.9)	23 (74.2)	60 (48.8)	0.011
Body/Tail		71 (46.1)	8 (25.8)	63 (51.2)	
Stage					
Resectable		12 (7.8)	3 (9.7)	9 (7.3)	0.449
Borderline Resectable		24 (15.6)	2 (6.5)	22 (17.9)	
Locally advanced		55 (35.7)	12 (38.7)	43 (35.0)	
Metastatic		63 (40.9)	14 (45.2)	49 (39.8)	
Laboratory characteristics					
WBC per μL		6860.0 (5565.0–8192.5)	7590.0 (5590.0–8660.0)	6810.0 (5550.0–7920.0)	0.212
Neutrophils per μL		4200.0 (3172.5–5422.5)	4600.0 (3390.0–6490.0)	4030.0 (3150.0–5200.0)	0.103
Lymphocytes per μL		1630.0 (1310.0–2040.0)	1600.0 (1140.0–2030.0)	1630.0 (1310.0–2060.0)	0.442
NLR		2.4 (1.8–3.6)	2.6 (2.0–4.3)	2.3 (1.7–3.3)	0.159
Hemoglobin, g/dL		12.6 (11.6–13.6)	12.6 (11.2–13.2)	12.5 (11.6–13.7)	0.429
Platelets, 10^3^/μL		242.5 (198.8–316.6)	273.0 (220.0–346.0)	228.0 (196.0–308.0)	0.033
Total bilirubin, mg/dL		0.6 (0.5–1.0)	0.8 (0.6–2.0)	0.6 (0.4–0.9)	0.002
AST, IU/L		20.0 (16.0–34.0)	25.0 (15.0–46.0)	20.0 (16.0–30.0)	0.168
ALT, IU/L		21.0 (14.8–40.5)	27.0 (17.0–48.0)	19.0 (14.0–38.0)	0.091
CA 19-9, U/mL		245.0 (28.5–1370.8)	611.0 (135.0–2364.0)	209.0 (20.8–1247.0)	0.160
Albumin, g/dL		4.1 (3.7–4.4)	4.0 (3.5–4.2)	4.1 (3.7–4.5)	0.151

Median (interquartile range) or *n* (%).

**Table 2 cancers-14-01244-t002:** Treatment characteristics preceding FN development.

Variables	All Patients (*n* = 154)	FN Group (*n* = 31)	Non-FN Group * (*n* = 123)	*p*-Value
Accumulation Dose, mg/m^2^				
5-FU	10,400.0 (9645.0–11,200.0)	8400.0 (5600.0–16,800.0)	10,500.0 (10,000.0–11,200.0)	0.237
Oxaliplatin	340.0 (303.9–340.0)	255.0 (170.0–510.0)	340.0 (340.0–340.0)	0.090
Irinotecan	720.0 (630.0–720.0)	540.0 (360.0–1080.0)	720.0 (675.0–720.0)	0.110

Median (interquartile range) * The median duration of FN development was the first four cycles. Because of this, cumulative doses in the non-FN group were calculated up to four cycles of chemotherapy. FN, febrile neutropenia; 5-FU, 5-fluorouracil.

**Table 3 cancers-14-01244-t003:** Univariate and multivariate analysis for the identification of the risk factors for febrile neutropenia.

		During All Cycles			
Variables		Univariate		Multivariate	
		OR (95% CI)	*p-*Value	OR (95% CI)	*p-*Value
Age, years					
≥65		2.30 (1.03–5.11)	0.042		
Sex					
Male		1.0	0.013	1.0	0.021
Female		2.77 (1.24–6.20)		2.87 (1.17–7.05)	
ECOG					
0		1.0	0.006	1.0	0.015
1		3.71 (1.45–9.46)		3.86 (1.31–11.42)	
BMI					
≥25		1.33 (0.55–3.20)	0.530		
Tumor location					
Body/tail		1.0	0.014	1.0	0.023
Head		3.02 (1.25–7.27)		3.06 (1.16–8.02)	
Stage					
Resectable		1.0			
Borderline resectable		0.27 (0.04–1.92)	0.192		
Locally advanced		0.84 (0.20–3.59)	0.811		
Metastatic		0.86 (0.20–3.60)	0.833		
NLR	<3	1.0	0.222		
	≥3	1.65 (0.74–3.67)			
Hemoglobin, g/dL	≥12	1.0	0.471		
	<12	1.34 (0.60–3.00)			
Total bilirubin, mg/dL	≤1.8	1.0	0.012		
	>1.8	3.46 (1.32–9.09)			
CA 19-9, U/mL	<1000	1.0	0.151		
	≥1000	1.82 (0.81–4.10)			
*UGT1A1* phenotype					
Extensive metabolizer		1.0		1.0	
Intermediate metabolizer		6.16 (2.16–17.54)	0.001	4.78 (1.61–14.21)	0.005
Poor metabolizer		5.50 (1.38–21.95)	0.016	4.86 (1.12–21.17)	0.035

BMI, body mass index; CA, carbohydrate antigen; CI, confidence interval; ECOG PS, Eastern Cooperative Oncology Group performance status; NLR, neutrophil to lymphocyte ratio; OR, odds ratio.

**Table 4 cancers-14-01244-t004:** Multivariate analysis using Cox regression analysis to identify risk factors for febrile neutropenia.

Variable	Febrile Neutropenia				
	Unadjusted HR	*p-*Value	Adjusted HR	95% CI	*p-*Value
Female sex	2.40	0.016	2.20	1.07–4.51	0.031
ECOG PS = 1	3.29	0.002	2.83	1.32–6.10	0.008
*UGT1A1* IM	5.15	0.001	4.30	1.61–11.52	0.004
*UGT1A1* PM	4.49	0.018	4.03	1.16–14.01	0.028

CI, confidence interval; ECOG PS, Eastern Cooperative Oncology Group performance status; HR, hazard ratio; IM, intermediate metabolizer; PM, poor metabolizer; *UGT1A1*, uridine diphospho-glucuronosyltransferase 1A1.

**Table 5 cancers-14-01244-t005:** Univariate and multivariate analysis related to grade 4 neutropenia.

Variable		During All Cycles			
		Univariate		Multivariate	
		OR (95% CI)	*p-*Value	OR (95% CI)	*p-*Value
Age, years					
≥65		1.95 (0.96–3.94)	0.063		
Sex					
Male		1.0	0.001	1.0	0.001
Female		3.47 (1.66–7.23)		3.44 (1.63–7.25)	
ECOG					
0		1.0	0.767		
1		0.88 (0.63–2.12)			
BMI					
≥25		0.76 (0.53–2.40)	1.127		
Tumor location					
Body/tail		1.0	0.152		
Head		1.61 (0.84–3.10)			
Stage					
Resectable		1.0			
Borderline resectable		1.66 (0.41–6.71)	0.481		
Locally advanced		3.13 (0.87–11.28)	0.081		
Metastatic		2.13 (0.61–7.46)	0.238		
NLR	<3	1.0	0.376		
	≥3	0.74 (0.38–1.45)			
Hemoglobin, g/dL	≥12	1.0	0.950		
	<12	0.98 (0.50–1.92)			
Total bilirubin, mg/dL	≤1.8	1.0	0.787		
	>1.8	1.14 (0.45–2.90)			
CA 19-9, U/mL	<1000	1.0	0.188		
	≥1000	1.62 (0.79–3.34)			
*UGT1A1* phenotype					
Extensive metabolizer		1.0		1.0	
Intermediate metabolizer		1.52 (0.77–3.01)	0.231	1.41 (0.69–2.88)	0.343
Poor metabolizer		4.05 (1.07–15.34)	0.039	4.06 (1.04–15.87)	0.044

BMI, body mass index; CI, confidence interval; ECOG PS, Eastern Cooperative Oncology Group performance status; NLR, neutrophil to lymphocyte ratio; OR, odds ratio.

**Table 6 cancers-14-01244-t006:** Multivariate analysis using Cox regression analysis to identify risk factors for grade 4 neutropenia.

Variable	Grade 4 Neutropenia				
	Unadjusted HR	*p-*Value	Adjusted HR	95% CI	*p-*Value
Female sex	2.06	0.001	1.98	1.31–2.98	0.001
*UGT1A1* IM	1.32	0.217	1.29	0.83–2.00	0.264
*UGT1A1* PM	2.01	0.026	1.79	0.96–3.33	0.066

CI, confidence interval; HR, hazard ratio; IM, intermediate metabolizer; PM, poor metabolizer; *UGT1A1*, uridine diphospho-glucuronosyltransferase 1A1.

**Table 7 cancers-14-01244-t007:** Serious hematologic toxicity (Grade III/IV) according to *UGT1A1* phenotype.

	Extensive Metabolizer (*n* = 71)	Intermediate Metabolizer (*n* = 66)	Poor Metabolizer (*n* = 17)	All Patients (*n* = 154)	*p-*Value
Neutropenia	57/71 (80.3)	58/66 (87.9)	15/17 (88.2)	130/154 (84.4)	0.411
Neutropenia (Grade IV)	38/71 (53.5)	42/66 (63.6)	14/17 (82.4)	94/154 (61.0)	0.082
Febrile neutropenia	5/71 (7.0)	21/66 (31.8)	5/17 (29.4)	31/154 (20.1)	0.001
Anemia	3/71 (4.2)	13/66 (19.7)	4/17 (23.5)	20/154 (13.0)	0.011
Thrombocytopenia	4/71 (5.6)	10/66 (15.2)	3/17 (17.6)	17/154 (11.0)	0.118

Data are presented as number of patients/total number (%).

**Table 8 cancers-14-01244-t008:** Incidence of diarrhea according to *UGT1A1* phenotype.

	Extensive Metabolizer (*n* = 71)	Intermediate Metabolizer (*n* = 66)	Poor Metabolizer (*n* = 17)	All Patients (*n* = 154)	*p-*Value
Diarrhea	22/71 (31.0)	21/66 (31.8)	4/17 (23.5)	47/154 (30.5)	0.801
Diarrhea (Grade III/ IV)	2/71 (2.8)	5/66 (7.6)	1/17 (5.9)	8/154 (5.2)	0.427

Data are presented as number of patients/total number (%).

## Data Availability

The data presented in this study are available in this article.

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
