# Peer review of "Impact of UGT1A1 Polymorphisms on Febrile Neutropenia in Pancreatic Cancer Patients Receiving FOLFIRINOX: A Single-Center Cohort Study"

_cancers, 2022, doi:10.3390/cancers14051244_

Round 1

Reviewer 1 Report

I examined the revised version of the manuscript entitled “Impact of UGT1A1 polymorphisms on febrile neutropenia in pancreatic cancer patients receiving FOLFIRINOX: a single-center cohort study”, in which Jiyoung Keum and colleagues investigated risk factors for febrile neutropenia (FN), in particular UGT1A1 polymorphisms, in a cohort of pancreatic cancer (PC) patients receiving FOLFIRINOX.

I wish to congratulate again the authors for their work, and for thoroughly addressing all the issues I highlighted in my previous comments. Considering the relevance of the subject and the lacking of proper literature, it should be encouraged to investigate these topic in further studies.

Concerning this paper, I have only a minor observation to report regarding statistical methods: authors decided to include intermediate UGT1A1 metabolizers (IM) in their multivariate analysis, although the univariate did not show any statistical significance (p value: 0.231) . In my opinion, it should be better discussed and argued the reason of this choice.

Author Response

We appreciate reviewer’s keen comment. We decided to include intermediate UGT1A1 metabolizers (IM) in the multivariate analysis, because IMs are known to be related to neutropenia, based on previous studies [1,2]. So, we added the following sentence in the result section as follows: “We included UGT1A1 IM group in the multivariate analysis in spite of statistical insignificance (p = 0.231) because previous studies showed that IM group was related to neutropenia.”

*"Reference*

1.de Man, F.M.; Goey, A.K.L.; van Schaik, R.H.N.; Mathijssen, R.H.J.; Bins, S. Individualization of irinotecan treatment: A review of pharmacokinetics, pharmacodynamics, and pharmacogenetics. /Clin Pharmacokinet /*2018*, /57/, 1229-1254.

2.Cheng, L.; Li, M.; Hu, J.; Ren, W.; Xie, L.; Sun, Z.P.; Liu, B.R.; Xu, G.X.; Dong, X.L.; Qian, X.P. Ugt1a1*6 polymorphisms are correlated with irinotecan-induced toxicity: A system review and meta-analysis in asians. /Cancer Chemother Pharmacol /*2014*, /73/, 551-560."

Reviewer 2 Report

Accept in present form 

Author Response

We appreciate reviewer’s comment. 

This manuscript is a resubmission of an earlier submission. The following is a list of the peer review reports and author responses from that submission.

Round 1

Reviewer 1 Report

In the manuscript entitled “Impact of UGT1A1 polymorphisms on febrile neutropenia in pancreatic cancer patients receiving FOLFIRINOX: a single-center cohort study”, Jiyoung Keumand colleagues aim to investigated risk factors for febrile neutropenia (FN), in particular UGT1A1 polymorphisms, in pancreatic cancer (PC) patients receiving FOLFIRINOX using a single-center cohort registry.

 The methodology of this study is based on a retrospective analysis of all patients diagnosed with pancreatic cancer treated at Severance Hospital since 2015 treated with full dose FOLFIRINOX without prophylactic G-CSF, available UGT1A1 genotype, measurable/evaluable disease, ECOG PS 0-1, adequate organ function and no prior anticancer treatment. According to these criteria, they identified 154 pts, 31 in the FN group and 123 in the non-FN group

Patients were divided into 2 cohorts: FN and non-FN. The authors propose UGT1A1 as the strongest predictive factor for FN and suggest UGT1A1 screening prior to chemotherapy.

I congratulate the authors for their manuscript, as it is indeed a very interesting topic. While there are growing data for increased risk for neutropenia and diarrhea following treatment with irinotecan in pts who are homozygous for the UGT1A1*28 allele, routine preemptive pharmacogenetic studies are not widely accepted and all new evidence to deepen the knowledge is welcome.

However, I consider relevant to specify some comments regarding this paper:

  1. it may be useful also to assess the impact of UGT1A1 polymorphisms on diarrhea in the study population, as it is the other great toxicity known to be influenced by UGT1A1
  2. the authors don’t mention the fact that other polymorphisms, such as those of (dihydropyrimidine dehydrogenase) DPD, could have an impact on FOLFIRINOX toxicity. A complete analysis of pharmacogenetic in pts treated with FOLFIRINOX should, in my opinion, include DPD to discriminate the impact of different drugs and polymorphisms in the toxicity.
  3. In the text, the authors specify that there were 55 pts affected by locally advanced and 63 were affected by metastatic disease, it should also be specified the status disease of the other 36 pts, that can be estrapolated only from the table.
  4. The inclusion criteria asked for ECOG PS ≤1, but then the results talk about ECOG PS ≥1. Were there pts with ECOG > 1, notwithstanding the inclusion criteria? If not, the text should be corrected
  5. It’s not specified whether pts had surgery for pancreatic cancer. As recent surgery is a relevant risk factor for FN and the study includes resectable and borderline resectable pts it could be useful to consider this information
  6. the authors don’t mention that, even if the routine preemptive use of this assay is not standard everywhere, testing for the presence of the UGT1A1*28 allele is available, and the US Food and Drug Administration (FDA)-approved label recommends testing since 2005, with reduced initial irinotecan doses in those who are homozygous for UGT1A1*28 to reduce the likelihood of dose-limiting neutropenia.

Author Response

Reviewer #1

Comments and Suggestions for Authors

In the manuscript entitled “Impact of UGT1A1 polymorphisms on febrile neutropenia in pancreatic cancer patients receiving FOLFIRINOX: a single-center cohort study”, Jiyoung Keum nd colleagues aim to investigated risk factors for febrile neutropenia (FN), in particular UGT1A1 polymorphisms, in pancreatic cancer (PC) patients receiving FOLFIRINOX using a single-center cohort registry.

The methodology of this study is based on a retrospective analysis of all patients diagnosed with pancreatic cancer treated at Severance Hospital since 2015 treated with full dose FOLFIRINOX without prophylactic G-CSF, available UGT1A1 genotype, measurable/evaluable disease, ECOG PS 0-1, adequate organ function and no prior anticancer treatment. According to these criteria, they identified 154 pts, 31 in the FN group and 123 in the non-FN group

Patients were divided into 2 cohorts: FN and non-FN. The authors propose UGT1A1 as the strongest predictive factor for FN and suggest UGT1A1 screening prior to chemotherapy.

I congratulate the authors for their manuscript, as it is indeed a very interesting topic. While there are growing data for increased risk for neutropenia and diarrhea following treatment with irinotecan in pts who are homozygous for the UGT1A1*28 allele, routine preemptive

pharmacogenetic studies are not widely accepted and all new evidence to deepen the knowledge is welcome.

However, I consider relevant to specify some comments regarding this paper:

  1. it may be useful also to assess the impact of UGT1A1 polymorphisms on diarrhea in the study population, as it is the other great toxicity known to be influenced by UGT1A1

Reply: We appreciate reviewer’s keen comment. We further analyzed the impact of UGT1A1 polymorphisms on diarrhea in Table 8. Neither diarrhea nor severe diarrhea (Grade III-IV) showed a significant association with the UGT1A1 polymorphism. These results differed from previous studies reporting the result of more diarrhea in UGT1A1 IM (intermediate metabolizer) and PM (poor metabolizer) groups than EM (extensive metabolizer) group, probably due to the small sample size. Retrospective analysis could also influence this outcome. We added these sentences to the result and discussion section.

  1. the authors don’t mention the fact that other polymorphisms, such as those of (dihydropyrimidine dehydrogenase) DPD, could have an impact on FOLFIRINOX toxicity. A complete analysis of pharmacogenetic in pts treated with FOLFIRINOX should, in my opinion, include DPD to discriminate the impact of different drugs and polymorphisms in the toxicity.

Reply: We appreciate reviewer’s keen comment. Due to the limitations of the retrospective study design, further analysis of other polymorphisms, including DPD, could not be carried out. We mentioned that incomplete analysis without DPD (dihydropyrimidine dehydrogenase) was a limitation of the study and added to the discussion section as follows:

“With respect to fluorouracil, DPD, TYMS, DPYS, CDA, and MTHFR have previously been reported to be associated with toxicity. It should be considered that these polymorphisms exist at lower frequencies among Asians, and the patient population in this study consisted of East Asians. [1-3]”

  1. In the text, the authors specify that there were 55 pts affected by locally advanced and 63 were affected by metastatic disease, it should also be specified the status disease of the other 36 pts, that can be estrapolated only from the table.

Reply: To address your comment, we clarified the cancer status of study subjects as follows:

“A total of 55 of the 154 patients (35.7%) had locally advanced PC, and 63 of the 154 (40.9%) had metastatic PC at diagnosis. The other 36 of the 154 patients (23.4%) were composed of 12 patients (7.8%) with resectable PC, and 24 patients (15.6%) with borderline resectable PC. Above them, 8 (66.7%) patients of resectable PC, and 16 (66.7%) patients of borderline resectable PC were received surgical resection for curative aim.”

  1. The inclusion criteria asked for ECOG PS ≤1, but then the results talk about ECOG PS ≥1. Were there pts with ECOG > 1, notwithstanding the inclusion criteria? If not, the text should be corrected

Reply: We appreciate reviewer’s keen comment. We revised what was indicated as “ECOG PS ≥1” to “ECOG PS = 1” in the text.

  1. It’s not specified whether pts had surgery for pancreatic cancer. As recent surgery is a relevant risk factor for FN and the study includes resectable and borderline resectable pts it could be useful to consider this information

Reply: We appreciate reviewer’s keen comment. We tried to analyze the association of recent surgery (≤ 1 month before chemotherapy start) and febrile neutropenia. However, of 24 patients who underwent surgery, only one of them was indicated as recent surgery, so analysis could not be performed. Recently, FOLFIRINOX has been administered as adjuvant chemotherapy for PC patients after surgery, so further research on the association between surgery and FN is needed. We added these sentences to the discussion section.

  1. the authors don’t mention that, even if the routine preemptive use of this assay is not standard everywhere, testing for the presence of the UGT1A1*28 allele is available, and the US Food and Drug Administration (FDA)-approved label recommends testing

since 2005, with reduced initial irinotecan doses in those who are homozygous for UGT1A1*28 to reduce the likelihood of dose-limiting neutropenia.

Reply: We appreciate reviewer’s keen comment. We mentioned about the FDA product label for irinotecan in the introduction section, but to make it clearer, we revised the sentence as follows:

“The US Food and Drug Administration (FDA)-approved product label for irinotecan recommends a reduced dose for patients homozygous for the UGT1A1*28 allele; however, the specific dose reduction for this patient population is not described. Because of uncertainty about clinical usefulness, preemptive UGT1A1 test is not widely accepted.”

Reviewer 2 Report

In this clinical study, authors have investigated the impact of UGT1A1 polymorphisms on febrile neutropenia in pancreatic cancer patients treated with FOLFIRINOX. One of the drugs Irinotecan in FOlFIRNOX combination is metabolized to an active metabolite through this enzyme, which can affect neutrophils. This is well done and well -presented study, however, major concern is novelty of the study

The role of UGT1A1 in context of FOLFIRINOX induced neutropenia has been explored in other cancers. This study confirms those findings in pancreatic cancer.

Concentrations of irinotecan metabolite influence neutrophil survival. Metabolism can be impacted by polymorphism. This is not expected to be influenced by the cancer type.

Author Response

Reviewer #2

Comments and Suggestions for Authors

In this clinical study, authors have investigated the impact of UGT1A1 polymorphisms on febrile neutropenia in pancreatic cancer patients treated with FOLFIRINOX. One of the drugs Irinotecan in FOlFIRNOX combination is metabolized to an active metabolite through this enzyme, which can affect neutrophils. This is well done and well -presented study, however, major concern is novelty of the study

The role of UGT1A1 in context of FOLFIRINOX induced neutropenia has been explored in other cancers. This study confirms those findings in pancreatic cancer.

Concentrations of irinotecan metabolite influence neutrophil survival. Metabolism can be impacted by polymorphism. This is not expected to be influenced by the cancer type.

Reply: We appreciate reviewer’s keen comment. There were two recommended regimens as first-line treatment for metastatic pancreatic cancer (PC). FOLFIRINOX and gemcitabine with albumin-bound paclitaxel (GNP). However, there was no established guideline for deciding the regimen.

If pancreatic mass occurs in the head of the pancreas, it can interfere with the excretion of the bile juice by obstructing the biliary tract, which affects drug metabolism. And biliary stent insertion for relieving biliary obstruction can cause inflammation and develop febrile neutropenia. The locational characteristics of pancreatic cancer, which can be susceptible to liver dysfunction and infection, may make it more vulnerable to the side effects of chemotherapy. Therefore, identifying the risk factor for FOLFIIRNOX associated febrile neutropenia in PC patients is necessary.

Reference

  1. Loh, M.; Chua, D.; Yao, Y.; Soo, R.A.; Garrett, K.; Zeps, N.; Platell, C.; Minamoto, T.; Kawakami, K.; Iacopetta, B. et al. Can population differences in chemotherapy outcomes be inferred from differences in pharmacogenetic frequencies? Pharmacogenomics J 2013, 13, 423-429.
  2. Rosmarin, D.; Palles, C.; Church, D.; Domingo, E.; Jones, A.; Johnstone, E.; Wang, H.; Love, S.; Julier, P.; Scudder, C. et al. Genetic markers of toxicity from capecitabine and other fluorouracil-based regimens: Investigation in the quasar2 study, systematic review, and meta-analysis. J Clin Oncol 2014, 32, 1031-1039.
  3. Yap, Y.S.; Kwok, L.L.; Syn, N.; Chay, W.Y.; Chia, J.W.K.; Tham, C.K.; Wong, N.S.; Lo, S.K.; Dent, R.A.; Tan, S. et al. Predictors of hand-foot syndrome and pyridoxine for prevention of capecitabine-induced hand-foot syndrome: A randomized clinical trial. JAMA Oncol 2017, 3, 1538-1545.